# Antibacterial Effect of Zinc Oxide-Based Nanomaterials on Environmental Biodeteriogens Affecting Historical Buildings

**DOI:** 10.3390/nano10020335

**Published:** 2020-02-16

**Authors:** Emily Schifano, Domenico Cavallini, Giovanni De Bellis, Maria Paola Bracciale, Anna Candida Felici, Maria Laura Santarelli, Maria Sabrina Sarto, Daniela Uccelletti

**Affiliations:** 1Department of Biology and Biotechnology “Charles Darwin”, Sapienza University of Rome, Piazzale Aldo Moro 5, 00185 Rome, Italy; emily.schifano@uniroma1.it; 2Department of Astronautical, Electrical and Energy Engineering, Sapienza University of Rome, Via Eudossiana 18, 00184 Rome, Italy; domenico.cavallini@uniroma1.it (D.C.); giovanni.debellis@uniroma1.it (G.D.B.); mariasabrina.sarto@uniroma1.it (M.S.S.); 3Department of Chemical Engineering Materials and Environment, Sapienza University of Rome, Via Eudossiana 18, 00184 Rome, Italy; mariapaola.bracciale@uniroma1.it (M.P.B.); marialaura.santarelli@uniroma1.it (M.L.S.); 4Department of Basic Applied Sciences for Engineering, Sapienza University of Rome, 00185 Rome, Italy; annac.felici@uniroma1.it

**Keywords:** stones, biodegradation, antimicrobial, cultural heritage, ZNGs

## Abstract

The colonization of microorganisms and their subsequent interaction with stone substrates under different environmental conditions encourage deterioration of materials by multiple mechanisms resulting in changes in the original color, appearance and durability. One of the emerging alternatives to remedy biodeterioration is nanotechnology, thanks to nanoparticle properties such as small size, no-toxicity, high photo-reactivity, and low impact on the environment. This study highlighted the effects of ZnO-based nanomaterials of two bacteria *genera* isolated from the Temple of Concordia (Agrigento’s Valley of the Temples in Sicily, Italy) that are involved in biodeterioration processes. The antimicrobial activities of ZnO-nanorods (Zn-NRs) and graphene nanoplatelets decorated with Zn-NRs (ZNGs) were evaluated against the Gram positive *Arthrobacter aurescens* and two isolates of the Gram negative *Achromobacter spanius*. ZNGs demonstrated high antibacterial and antibiofilm activities on several substrates such as stones with different porosity. In the case of ZNGs, a marked time- and dose-dependent bactericidal effect was highlighted against all bacterial species. Therefore, these nanomaterials represent a promising tool for developing biocompatible materials that can be exploited for the conservation of cultural heritage. These nanostructures can be successfully applied without releasing toxic compounds, thus spreading their usability.

## 1. Introduction

The biodeterioration process refers to any form of irreversible alteration and any undesirable change in the properties of materials, caused by the activity of living organisms. It is known that microbial activities are one of the main factors of the degradation of cultural heritage that, together with other causes such as climatic factors, contribute to the deterioration of inorganic substrates, leading to physical–chemical and aesthetic damages. Indeed, changes in temperature, humidity, rain, erosion, wind or atmospheric pollution induce the growth of microbial biofilms and hence the deterioration processes [1,2].

Microorganisms grow on different substrates, altering the original color and appearance, thanks to their diversified metabolic activities and release of pigments. Bacteria biofilm can colonize organic and inorganic materials using them as chemical and energy sources and causing extensive and irreversible damage to artifacts of any type or composition, such as stones, frescoes, books, or canvas [3,4,5]. Hence, the type of substrate on which the microorganisms settle, the availability of nutrients, the mineralogical composition, and water permeability [6], often determine the mechanism of biodeterioration.

Organisms that irreversibly alter the substrates on which they develop are defined as biodeteriogens, and they include bacteria, cyanobacteria and algae (considered to be pioneers in the biofilm formation on stone materials), fungi, and higher plants [7,8].

This biological colonization, also known as “biofouling”, has been the subject of several studies since the 1980s [9]. These biological processes cause much long-term damage, and, for this reason, a number of methods have been developed to control the growth of microorganisms and to avoid biodeterioration on monuments with high archaeological or cultural heritage value. For example, synthetic chemicals, aqueous leaf extracts from plants, and potent ionic antimicrobial agents based on metals were tested [10,11,12]. One of the emerging alternatives is nanotechnology, which seems to be promising in numerous fields such as medicine, agriculture, and the environment [13].

In recent years, nanotechnology has become commonly used in restoration and consolidation of historical monuments, because of the unique features of nanoparticles and nanostructures [14]. These, including physical, optical, and magnetic properties, allow the possibility to realize transparent hydrophobic and bactericidal protective layers on surfaces. In early studies, nanomaterials were designed to improve the conservation and restoration of the damage to historical and cultural heritage, such as mural paintings and stone buildings [15,16]. In the last decade the antimicrobial effects of metal nanoparticles were explored and reported in many works [17,18,19]. Furthermore, one of the most important features of nanoparticles is their photocatalytic ability that avoids biofilm formation of microorganisms on historical buildings and allows easy removal of dirt and spots from the surfaces [20,21,22]. The self-cleaning and antimicrobial effect of nanoparticles is therefore a remedy against air pollution as well as contaminants. In particular, zinc oxide (ZnO) is a non-toxic semiconductor, which, thanks to its properties, can form nanostructures with different morphologies [23,24]. It has been shown that ZnO nanorods (ZnO-NRs) can inhibit the growth of bacteria such as *Streptococcus mutans* by acting as nanoneedles [25]. They have also been shown to be effective towards other pathogenic bacteria such as *Staphylococcus aureus* [26]. Another class of nanomaterials showing to have antimicrobial power is the ZnO-NRs-decorated GNPs (ZNGs) in which pristine graphene nanoplatelets (GNPs) are decorated with ZnO-NRs through a simple hydrothermal method [27,28].

The objective of this research was to evaluate the antimicrobial activity and antibiofilm formation of ZnO-NRs and ZNGs against bacterial species, isolated from the Temple of Concordia (Agrigento’s Valley of the Temples in Sicily, Italy), responsible for the biodeterioration phenomena. The exploitation of these nanomaterials for the preservation of stones with different porosity was also investigated.

## 2. Materials and Methods

### 2.1. Production of Nanostructures and Suspensions

Zinc oxide nanorods (ZnO-NRs) were synthesized through thermal decomposition of zinc acetate di-hydrate and successive probe sonication as described in [29]. Their morphology was observed through high-resolution field emission scanning electron microscopy (FE-SEM). GNPs were produced by thermal expansion at 1050 °C for 30 s of commercially available Graphite Intercalation Compound (GIC), and successive liquid-phase exfoliation by probe sonication [25]. About the synthesis of ZnO-NR-decorated graphene nanoplatelets (ZNGs), rod shaped ZnO-NRs, with a diameter of ~36 nm and length in the range of 300–400 nm, were directly grown over the planar shaped pristine graphene nanoplatelets (GNPs), forming ZNGs [27]. Aqueous colloidal suspensions of ZNGs were prepared through the dispersion of ZNG powder in ultrapure and sterilized deionized water using probe ultrasonication. The homogenous suspensions were then readily transferred to 50 mL sterilized centrifuge tubes. The characterization of ZNGs was previously performed and is described in [25].

### 2.2. Bacterial Strains and Growth Conditions

Bacteria from the Temple of Concordia in Sicily were collected using sterile cotton swabs and then placed in a sterile tube containing 2 mL of Luria-Bertani (LB) and finally incubated in the laboratory at 30 °C for 24 h. The collected samples were then plated on LB agar plates and, after further incubation at 30 °C for 24 h, morphologically different colonies were purified. Bacterial strains isolated from the Temple of Concordia in Sicily and used in this study were *Achromobacter spanius* TC1, *Achromobacter spanius* TC7, and *Arthrobacter aurescens* TC4. For experiments, they were grown in LB broth at 30 °C overnight.

### 2.3. DNA Isolation and Sequencing

DNA was extracted and amplified according to [30]. A region of approximately 1400 bp from the 16S rRNA gene was amplified using the primers F8 (5′-AGAGTTTGATCCTGGCTCAG-3′) and R1492 (5′-GGTTACCTTGTTACGACTT-3′). The PCR reaction was performed utilizing the Taq DNA polymerase from Accuzyme DNA Polymerase (Bioline). BMR Genomics (Padova, Italy) sequenced the amplified region and the obtained sequences were analyzed with BLAST database.

### 2.4. Cell Viability Test

Viability in liquid assay was analyzed as described in [28]. Samples were incubated with minimal agitation at 30 °C, under diffuse visible light. After treatments, the aliquots of the different samples were diluted and then grown on agar-LB plates; after incubation at 30 °C overnight, the bacterial ability to form new colonies was measured by counting the number of colonies forming units (CFU).

### 2.5. Evaluation of Biofilm Formation

Biofilm formation was evaluated as previously described [28] with some differences: each well was filled with 200 µL of LB broth and 125 μg/mL of ZnO-NRs or ZNGs (in triplicate). Wells without nanoparticles were utilized as controls. Next, plates were incubated at 30 °C for 24 h under diffuse visible light and Crystal Violet assay was performed. The experiment was repeated three times with three replicates for each treatment.

### 2.6. LIVE/DEAD Assay

An amount of 1 × 10^7^ cells/mL was inoculated in LB in the presence of a ZnO-NRs concentration of 125 μg/mL or a ZNGs concentration of 10 μg/mL. The suspension was incubated in a 35 mm Petri plate with a cover slide for 24 h at 30 °C under diffuse visible light. As control, strains grown in LB were inoculated in the absence of nanoparticles. After treatment, cultures were gently removed, washed twice with sterile water, and stained using a LIVE/DEAD BacLight–Bacterial Viability (Kit 7012, Invitrogen, Mount Waverley, Australia) for 15 min at room temperature in the dark. After washing with sterile deionized water, stained cells were observed under a Zeiss Axiovert 25 fluorescence microscope. Syto9 is able to stain live and dead cells, while propidium iodide is unable to penetrate the intact cell wall in viable cells, staining only dead cells.

### 2.7. Preparation of Stones

Suspensions of ZNGs to be used with the stones were produced at 250 μg/mL through bath sonication (at 37 kHz of working frequency) of 10 mg of ZNGs, prepared as described before, in 40 mL deionized water for 3 min. Noto stone, Carrara marble and yellow brick specimens (1 × 1 × 0.5 cm^3^) were then spray coated, through an airbrush, with the aqueous suspension of ZNGs. Full coverage of the specimens was ensured by repeating the spraying process four times. Each time 2 mL of suspension was sprayed and evaluated by FE-SEM analysis.

### 2.8. Characterization of Stone Materials

Porosity and density of stone specimens were characterized through mercury porosimetry. The water capillarity absorption experiments were performed following the UNI EN 15801:2010 [31]. Colorimetric measurements were performed on stone samples to verify color modification due to ZNGs exposition according to [32]. The ΔE value refers to the median values of the treated samples compared to the untreated ones. Furthermore, according to the literature, ΔE < 5 was considered as corresponding to a not-significant variation [33]. The measurements were made on untreated and treated samples, on three different zones on stone surfaces.

### 2.9. Antimicrobial Activity of Treated Stones

Treated surfaces of the three different specimens were submerged in a concentration of 1.0 × 10^7^ cells/mL of bacterial cultures and then incubated at 30 °C for 24 h, under diffuse visible light. After that, the treated stones were dipped into 3 mL of sterile deionized water, shaken for 1 min and then, 1 mL was plated on LB plates. After incubation at 30 °C for 24 h the bacterial colonies were counted. Untreated stone samples were used as reference. The experiments were repeated three times with three replicates for each treatment.

### 2.10. FE-SEM Microscopy Imaging for Treated Stones

Biological samples were prepared according to the procedures described above. After 24 h of treatment, stones in the presence or not of bacteria were fixed with 2% glutaraldehyde in PBS for 1 h at RT in the dark. After 3 washes in PBS, samples were then dehydrated as described in [34]. The stone samples were successively utilized for the electron microscopy analysis.

### 2.11. Statistical Analysis

Experiments were performed at least in triplicate. Data are presented as mean ± SD, and Student’s-test or one-way ANOVA analysis coupled with a Bonferroni post-test (GraphPad Prism 4.0 software) was used to determine the statistical significance between experimental groups. Statistical significance was defined as * *p* < 0.05, ** *p* < 0.01, and *** *p* < 0.001.

## 3. Results and Discussion

### 3.1. Bacteria Identification and ZnO-NRs Cell Viability Assay

In this study, antimicrobial and antibiofilm properties of ZnO-based nanomaterials were evaluated against two bacterial *genera* involved in biodeterioration processes. Especially, several bacterial colonies were isolated from the Temple of Concordia in Sicily and identified at the molecular level by the amplification of 16S rDNA. The comparison of the obtained sequences with those in the BLAST database allowed identification of the Gram-positive *Arthrobacter aurescens*, and two strains belonging to the Gram-negative *Achromobacter spanius*, which are involved in biodeterioration, as has been reported in many works [35,36,37]. The three different strains were confronted with the ZnO-NR-based nanomaterials at different concentrations.

Initially, bacterial survival was evaluated by CFU counting analysis after incubation with ZnO-NRs in liquid assay. After 2 h of treatment, a significant reduction in cell viability was observed at concentrations of 50 µg/mL and 100 µg/mL, with respect to untreated cells. In the case of *A. spanius* TC7, a notable effect was highlighted also at lower concentration of 10 µg/mL. These results were marked in 24 h-exposure, where a mortality rate of over 98% was observed already with 10 μg/mL for all the strains (Figure 1).

### 3.2. Evaluation of ZnO-NRs Antibiofilm Properties

Biodeterioration of monuments is mainly due to the ability of microorganisms to create biofilm: in fact, bacteria can stick to each other and to a surface, forming a group of cells producing an extracellular matrix composed of DNA, proteins and polysaccharides. In these biofilms, microorganisms resist adverse abiotic conditions [2]. Consequently, cultural heritage decay problems are related to the biofilm formation and its metabolic activity, such as production of organic and inorganic acids and bacteria exopolymers [38].

Biofilms, grown on stone monuments, increase rock deterioration through physicochemical processes, as has been observed in many stone structures of cultural heritage [39,40]. For this reason, antibiofilm effects of ZnO-NRs against the three isolated microorganisms were tested at a concentration of 125 μg/mL. After 24 h of treatment, the biofilm formation on glass slides was evaluated by fluorescence microscope utilizing LIVE/DEAD assay (Figure 2). The Syto9 staining of *A. aurescens* TC4 and *A. spanius* TC7 strains showed a reduced number of treated cells on the surface (Figure 2a,c respectively). Instead, when the propidium iodide was used, no differences in the number of death cells were recorded comparing treated and untreated samples (data not shown).

These results suggest that the nanomaterial is able to inhibit the adhesion of these species, to the surface. Instead, in the case of *A. spanius* TC1 an increased number of treated cells stained with Syto9 was observed in comparison with the untreated counterpart, suggesting a stimulation of biofilm formation by the nanomaterial (Figure 2b). The biofilm on plastic surface was also quantified by the Crystal Violet method (Figure 3). In *A. aurescens* TC4 and *A. spanius* TC7 an inhibition of about 90% of biofilm was observed (Figure 3a,c). Conversely, ZnO-NRs seemed to stimulate almost 50% of biofilm forming capacity in *A. spanius* TC1 (Figure 3b). Those results demonstrated the effectiveness of ZnO-NRs in controlling biofilm growth of only two isolated microorganisms, involved in biodeterioration.

Surprisingly, *A. spanius* TC1 and *A. spanius* TC7 showed different behavior when treated with ZnO-NRs, highlighting the presence of intra-species differences between the two strains. Indeed, as reported in biodiversity studies of several bacterial species, intra-species analysis revealed extensive variation between isolates, both at the genotypic and phenotypic level [41,42].

### 3.3. ZNGs Antimicrobial and Antibiofilm Activity

An innovative nanomaterial consisting of graphene nanoplatelets decorated by zinc oxide nanorods (ZNGs) was described in previous work for the antimicrobial and antibiofilm properties (Zanni et al., 2017). Therefore, experiments were performed with ZNGs nanomaterial: cell viability test was performed with ZNGs at different concentrations (Appendix A). After just 2 h, ZNGs treatment (1 μg/mL) resulted in almost 50% antibacterial activity against *A. aurescens* TC4 and *A. spanius* TC7, and in 70% against *A. spanius* TC1 at a concentration of 10 μg/mL. These results show a time- and dose-dependent antimicrobial effect of ZNGs against the planktonic forms of two bacteria *genera*. After 24 h of treatment, an inhibition of bacteria growth of over 70% was observed with 1 μg/mL. Especially, when the LIVE/DEAD assay was performed with ZNGs, fluorescence images showed the ability of ZNGs to inhibit *A. spanius* TC1 biofilm already at a concentration of 10 μg/mL (Figure 4a). Similarly, *A. aurescens* TC4 and *A. spanius* TC7 biofilms were also inhibited (Appendix A). The Crystal Violet assay on plastic surface underlined a high inhibitory activity of ZNGs also against *A. spanius* TC1 biofilm formation with a decrease of 50% compared to the control (Figure 4b). In the case of *A. aurescens* TC4 and *A. spanius* TC7 the data obtained with ZNGs treatment were comparable with that obtained with ZnO-NRs (data not shown).

Results demonstrate that biofilm production was reduced by ZnO-based nanomaterial treatment, in agreement with the observation that bacteria in biofilms are more resistant to antibacterial agents than their planktonic form [43]. For the Gram-negative *A. spanius* TC1, ZNGs treatment was more efficient, even when lower concentrations were used. The effect of ZNGs on the biofilm of *A. spanius* TC1 can be related to a different cell wall composition with respect to *A. spanius* TC7, due to intra-species variation. Indeed, the Gram-negative *Chlamydia psittaci* isolated microorganisms showed variations in the putative outer membrane genes [44].

### 3.4. ZNGs Actions on Stones

A large percentage of the world’s cultural heritage is made from stone, which is disappearing, because of the different deterioration phenomena, one of which is biodeterioration. The most common stones are calcareous stones, such as marble and limestone, and siliceous ones, such as sandstone, granite, and artificial stones. They are different in hardness, porosity, and alkalinity, and their properties affect their predisposition to biodeterioration. In this study, three materials with different structural features, as reported in Table 1, were used to test the antimicrobial properties of ZNGs: common yellow brick, Noto stone, and Carrara marble.

The apical surfaces of the stone specimens were coated with 250 μg/mL ZNG-suspension, to cover the porous surfaces of the materials preventing the proliferation of the bacteria in depth. In order to evaluate if the porosity and pore size influenced the distribution and position of the nanostructures on the surface, an FE-SEM analysis of these materials was carried out. The micrographs of treated and untreated samples showed the different disposition of the nanostructures on surfaces with diverse pore size, highlighting the uniformity of ZNGs distribution on the less porous Carrara marble, unlike the other specimens having a higher porosity (Figure 5). Furthermore, colorimetric analysis showed no significant variation between treated and untreated stone (Appendix A).

The antibacterial activities of the treated stones against the Gram-positive *A. aurescens* TC4 and *A. spanius* TC1, representative of Gram-negative isolated microorganisms, are reported in Figure 6. Treated Noto stones, typical diffused stones in historical buildings in Sicily and Malta, revealed a reduction of viability up to 60% for both bacterial strains (Figure 6a). At the same time, treated common yellow brick induced a strong reduction of *A. aurescens* TC4 and *A. spanius* TC1 cell viability of about 90% (Figure 6b). A reduction of about 70% in bacterial viability was obtained in the case of treated Carrara marble (Figure 6c). Similar results were also obtained when *A. spanius* TC7 strain was utilized (data not shown).

Generally, these results indicated a correlation between the antibiofilm properties of ZNGs and nanostructure distribution, depending on the specimen’s porosity. Indeed, the high porosity of Noto stone causes a reduced distribution of ZNGs on the surface, leading to a slight reduction of the antibiofilm effect. In Figure 7, it is possible to observe, as an example, the Noto stone covered by ZNGs treated or not with *A. aurescens* TC4 cells (Panels a,b, respectively). The bacterial cell wall showed mechanical injuries caused by direct contact with the nanomaterial. In fact, ZnO-NRs function as nanoneedles that pierce the bacterial wall, while nanosheets of GNPs offer a large surface area for the oriented growth of the ZnO-NRs over the GNP surface.

Consequently, this behavior already confirmed what has been observed in previous work [25], namely that ZnO-NRs deposited on the GNPs surface provoke mechanical aggression on the cell walls. Especially, the use of graphene nanoplatelets (GNPs) creates an advantage, forcing the growth of the ZnO-NRs in a specific orientation and extending the area of their action. Indeed, this shape contributes to increasing the adhesion of the nanostructures to the cell wall enhancing the penetration of the ZnO-NRs through the cell membrane, as highlighted in FE-SEM analysis performed in the previous work [25]. Therefore, this may cause greater ability to damage the bacterial surface, through mechanical damage produced by the ZnO-NRs [45,46].

Furthermore, the use of ZNGs could bypass the treatment of biodeterioration with traditional biocides, obtaining a safer solution than toxic chemical agents. Indeed, it has been demonstrated that ZnO-NRs and ZNGs lacked acute toxicity in vitro and in vivo: ZNO-based nanomaterials did not affect viability, fertility, and body length in different model organisms [26,28,47].

Generally, several studies suggested that ZnO and graphene-based nanomaterials represent good candidates to be used for restoration of cultural heritage, thanks to their many advantages, such as low cost and effectiveness in size dependency against a wide range of bacteria [48,49]. Commonly, many biocides used in cultural heritage cover a wide range of chemical classes, such as inorganic compounds (in e.g., Na and Ca hypochlorite) and organic ones, like the quaternary ammonium compounds [50,51]. Most of the commercial products based on these chemical toxic compounds have been forbidden due to their environmental and health hazards [52]. Indeed, traditional chemical biocides are usually aggressive towards the damaged substrates as well as to workers’ health. The graphene-based nanomaterials, with their biocompatible properties, could represent an alternative to conventional treatments, to be applied as antimicrobial agents on stone monuments [53].

## 4. Conclusions

Our studies indicated that ZnO-based nanomaterials show antimicrobial and antibiofilm properties against different bacteria involved in stone biodeterioration. Indeed, results suggest that ZnO-NRs-decorated GNPs might be highly effective in preserving the surface of artwork in a cost-effective way that enables control of bacteria growth and therefore biodeterioration processes also at very low concentrations. These nanomaterials could be used to fight biodeterioration, offering the advantageous possibility of removing biofilm and limiting the consequent physical and chemical damages. ZNGs could avoid the use of traditional bioremediation reagents (such as biocide products) which otherwise, would require excessive and unfavorable chemical conditions. Furthermore, commercial chemical treatments can be hazardous for workers, the environment, and in some case for the artwork itself.

## Figures and Tables

**Figure 1 nanomaterials-10-00335-f001:**
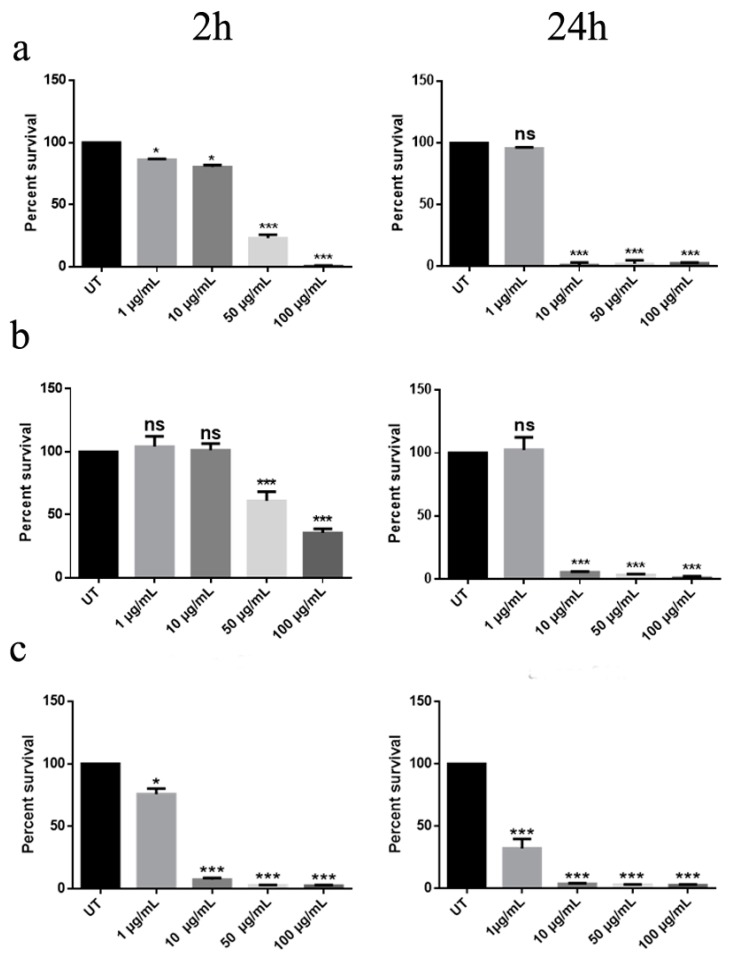
Effect of zinc oxide nanorods (ZnO-NRs) on different strains viability. Bacteria were treated or not (UT: untreated) with different concentrations of nanorods for 2 h (left column) or 24 h (right column) and bacterial survival was evaluated by CFU counting analysis. (**a**–**c**) indicate respectively *A. aurescens* TC4, *A. spanius* TC1, and *A. spanius* TC7. A one-way ANOVA analysis with the Bonferroni post-test was used to assess statistical significance (ns not significant; * *p* < 0.05 and *** *p* < 0.001 with respect to UT).

**Figure 2 nanomaterials-10-00335-f002:**
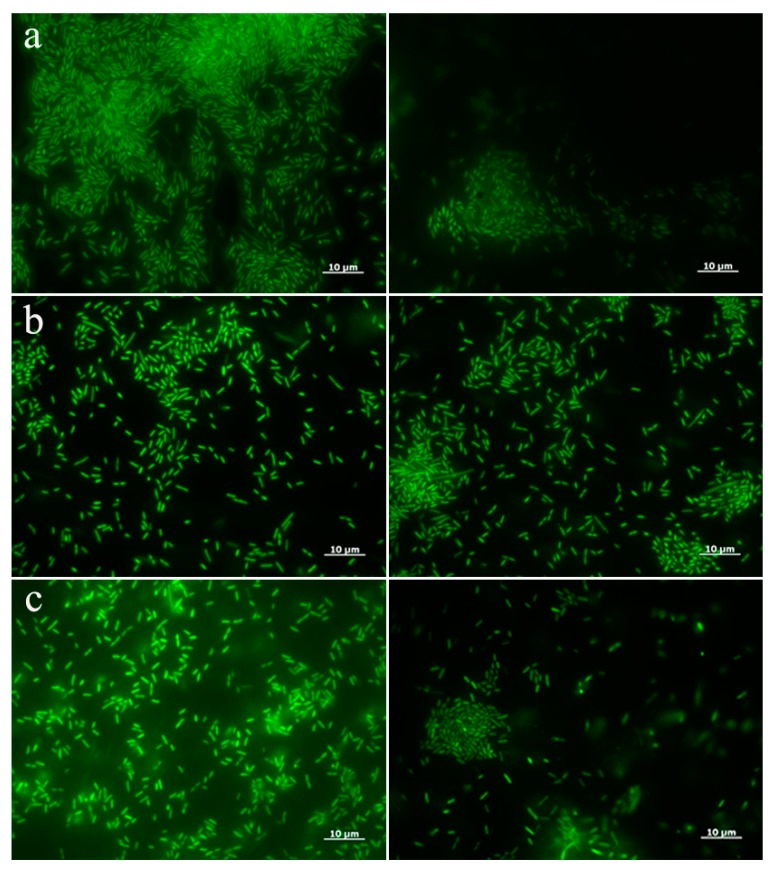
LIVE/DEAD staining after treatment with ZnO-NRs. Fluorescence microscope images of untreated bacterial strains (left) and treated cells with a concentration of ZnO-NRs of 125 μg/mL (right). (**a**–**c**) indicate respectively *A. aurescens* TC4, *A. spanius* TC1, and *A. spanius* TC7.

**Figure 3 nanomaterials-10-00335-f003:**
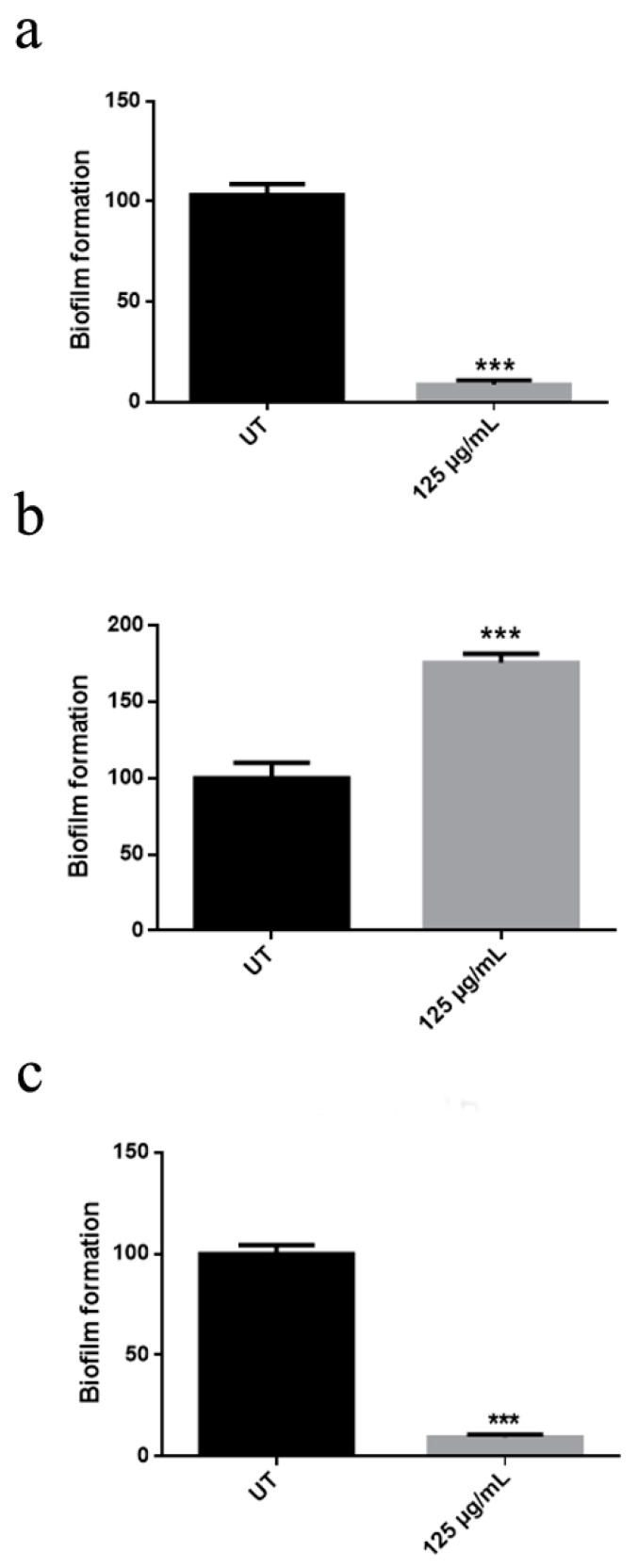
Effect of ZnO-NRs on biofilm formation. Different strains were grown in the absence (UT: untreated) or presence of 125 μg/mL ZnO-NRs. (**a**–**c**) indicate respectively *A. aurescens* TC4, *A. spanius* TC1, and *A. spanius* TC7, after 24 h of treatment. For statistical analysis one-way ANOVA method coupled with the Bonferroni post-test was used (*** *p* < 0.001 with respect to UT).

**Figure 4 nanomaterials-10-00335-f004:**
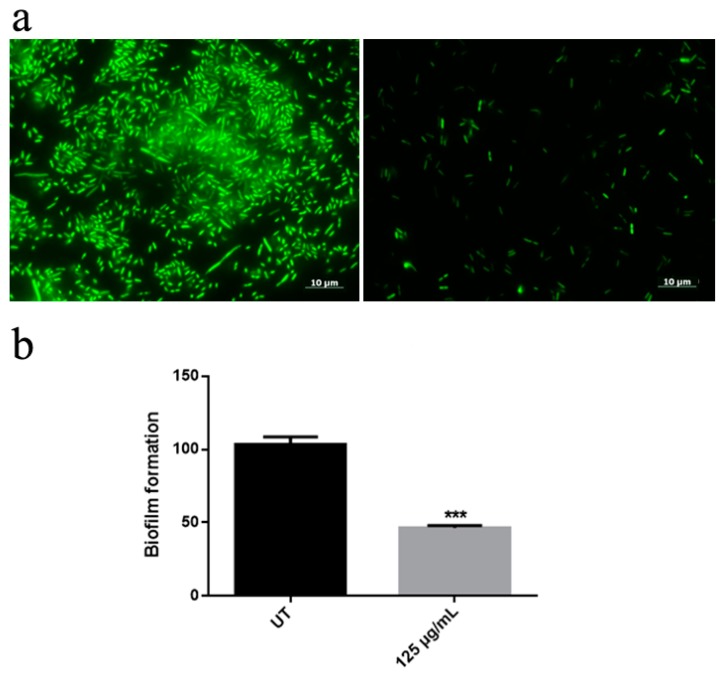
(**a**) LIVE/DEAD staining after *A. spanius* TC1 treatment with Zn-NRs (ZNGs). Fluorescence microscope images of untreated bacterial strains (on the left) and treated cells with a concentration of ZNGs of 10 μg/mL; (**b**) Effect of ZNGs on biofilm formation on *A. spanius* TC1. Bacteria were grown in the presence of 125 μg/mL ZNGs while samples without nanomaterials were taken as controls. To assess statistical analysis a one-way ANOVA analysis with the Bonferroni post-test was used (*** *p* < 0.001 with respect to UT).

**Figure 5 nanomaterials-10-00335-f005:**
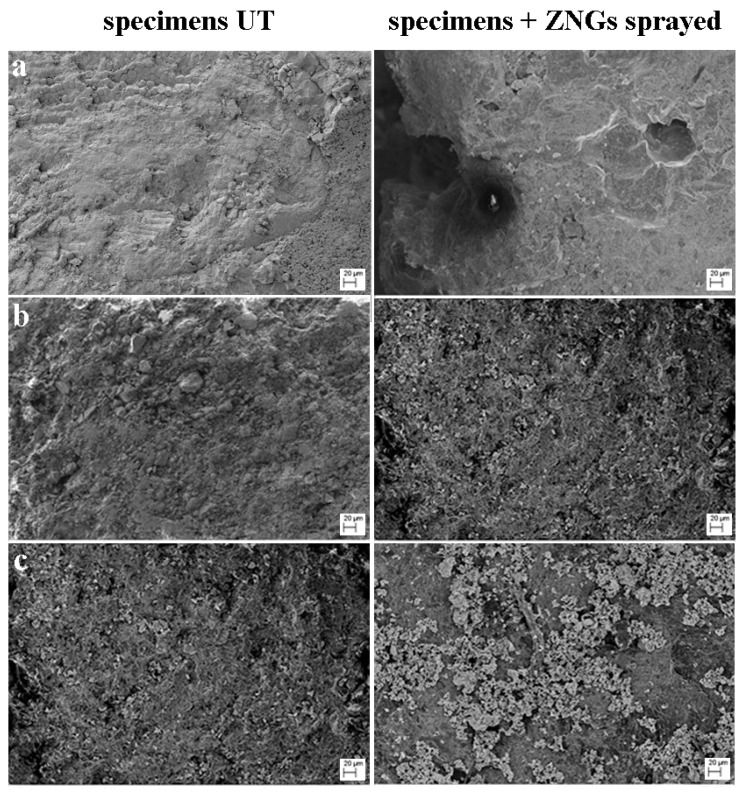
FE-SEM images of different untreated and sprayed with ZNGs 250 μg/mL specimens. (**a**–**c**) indicate Noto stone, Common yellow brick and Carrara marble, respectively.

**Figure 6 nanomaterials-10-00335-f006:**
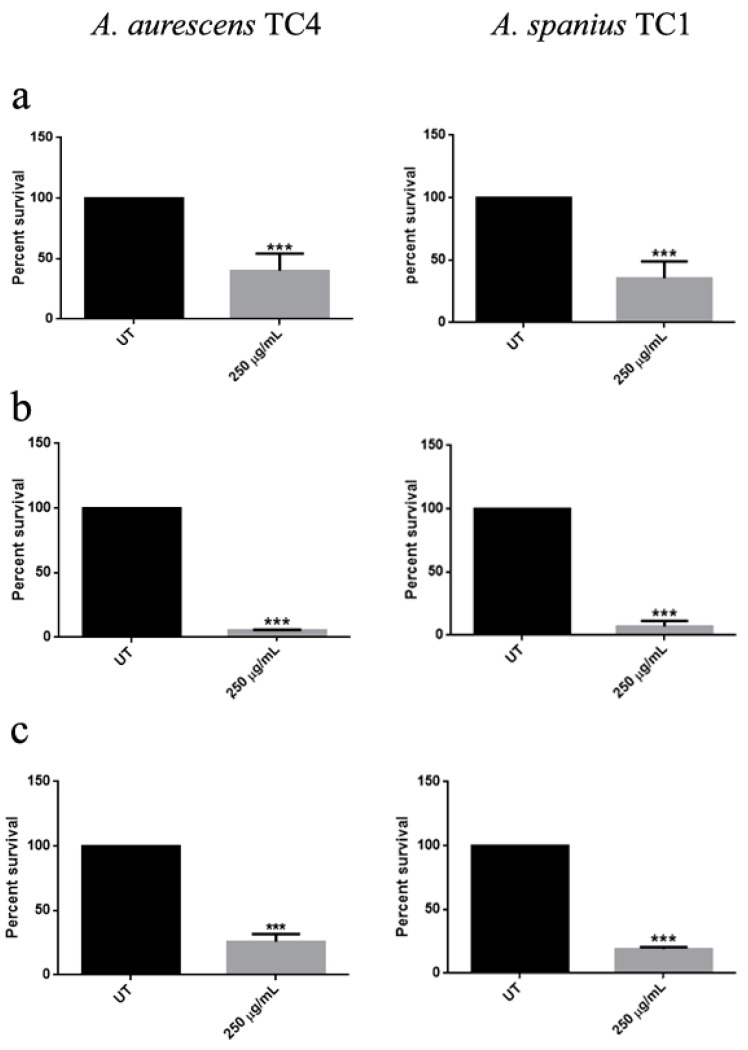
Cell viability after treatment of stone specimens with ZNGs. CFU counting of the indicated species after 24 h of incubation on treated stones. (**a**–**c**) indicate respectively Noto stone, common yellow brick, and Carrara marble. To assess statistical analysis a one-way ANOVA analysis with the Bonferroni post-test was used (*** *p* < 0.001 with respect to UT).

**Figure 7 nanomaterials-10-00335-f007:**
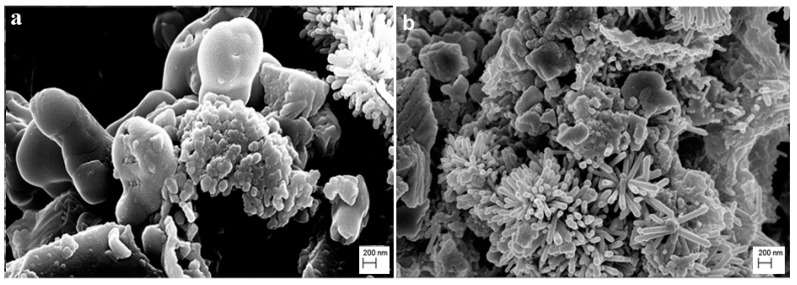
FE-SEM micrograph of *A. aurescens* TC4 cells after exposure to Noto stone covered with ZNGs. Panel (**a**), bacterial cell after 24 h exposure; Panel (**b**), brick covered by ZNGs. Bar, 200 nm.

**Table 1 nanomaterials-10-00335-t001:** Physical properties of the selected materials.

	Density (g/cm^3^)	Porosity (%)	Water Absorption (%)
Noto stone	1.65 ± 0.03	38.8 ± 0.04	12-21 ± 0.03
Carrara marble	2.7 ± 0.04	0.4 ± 0.07	0.11 ± 0.10
Common yellow brick	1.48 ± 0.05	28.5 ± 0.05	24 ± 0.08

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
