# Peer review of "Antibacterial Effect of Zinc Oxide-Based Nanomaterials on Environmental Biodeteriogens Affecting Historical Buildings"

_nanomaterials, 2020, doi:10.3390/nano10020335_

Round 1

Reviewer 1 Report

The authors would like to describe the use of previously reported nanomaterials, based on zinc oxide, to counteract the proliferation of certain microbes, which are known to prosperate on the stones of certain buildings of historical interest. In this sense, the originality of the work is limited, especially compared to the authors’ previous works; the only element of novelty would be the application for cultural heritage protection.

Nevertheless, it is not clear what is the reason to use zinc oxide together with graphene. What is the advantage brought by the use of this latter? The authors claim that it favors the generation of a certain morphology of zinc oxide particles with desirable mechanical properties. But, where are the characterization results? Important details are missing, including the accurate description of the “graphene  nanoplatelets” used. What is the isolation procedure used for the studied bacterial strains? What were the growth conditions (Petri dish, bottle, test-tube...). The lack of such details significantly impairs reproducibility, which is against the guidelines of this Journal.

Since the goal of this work is to demonstrate the antibacterial and antibiofilm properties of their zinc oxide-based products, the authors must clarify their actual mechanism of action. Zinc oxide, as correctly mentioned in the Introduction, is known to be photocatalytic and thus able to generate reactive species which, in turn, can significantly contribute to the antimicrobial activity. The authors must precise whether they performed their tests in the dark or in presence of daylight, diffuse or direct, or under the light of fluorescent lightbulbs. Experimental controls in this sense should be performed as well. This is especially important by taking into account the final application of the proposed products, that is, on buildings exposed to sunlight. Only then, their hypothesis about antimicrobial activity by mechanical damaging could be accepted, or amended by including a photocatalytic component.  

Why the authors decided not to show the results obtained with propidium iodide, that is, the dead cells? They are strongly encouraged to show them, at least in the Supporting Information.

Also: would the described products be significantly convenient (also from an economic perspective) compared to commercial ones? What about the possibility of deleterious effects on the stone themselves generated by side-reactions? A proper comparison is needed: the authors must extend the scarce discussion they provide at the end of the manuscript.

In general, the writing style should be improved. Especially in the Introduction and Discussion sections, there are some ingenuities, including the repeated use of expressions such as “nanoknives” and so on, which does not sound professional.

The authors should also consider to rewrite the final sections of the Abstract and of the Conclusions, since they give the impression that the hazards and limitations refer to the products they propose, rather than to usual ones.

In Figure 5a, the scale is clearly different from that of the other ones: 100 microns compared to 20 microns. This makes very difficult, if not impossible, to properly compare the pictures. Also, in the legend, the authors should indicate more clearly the nature of the treatment used: “sprayed with 250 μg/mL” of what?

Figure 7: the image of one bacteria flattened on a carpet of nanorods is not enough to provide convincing evidence for the “mechanical” antimicrobial mechanism. See before.

In general, by accompanying the SEM images with EDX maps would make the results more convincing, especially by showing the distribution of carbon (from bacteria) in samples treated with only zinc oxide.

It is also apparent, from the results showed, that the activity of the different materials tested is not the same for different bacteria strains: in other words, they cannot be considered for “general purpose”. This, in turn, means that they might favor the selection of one or more bacterial species, possibly exacerbating the stone degradation problems.

What is the durability of the proposed materials? That is, for how long they will retain their antimicrobial properties, especially under normal weathering conditions, before a new application would be necessary?

The authors must provide a satisfying explanation to all these points in order to make their work, which anyway remains a proof-of-concept (the authors must clarify this point) publishable in Nanomaterials.

Minor details:

Lines 120, 126, 142 and so on: instead of “H2Odd”, it is better to say “deionized water”.

Line 123: it is “propidium”.

Line 140: correct the exponent.

Author Response

The authors would like to describe the use of previously reported nanomaterials, based on zinc oxide, to counteract the proliferation of certain microbes, which are known to prosperate on the stones of certain buildings of historical interest. In this sense, the originality of the work is limited, especially compared to the authors’ previous works; the only element of novelty would be the application for cultural heritage protection.

Nevertheless, it is not clear what is the reason to use zinc oxide together with graphene. What is the advantage brought by the use of this latter? The authors claim that it favors the generation of a certain morphology of zinc oxide particles with desirable mechanical properties. But, where are the characterization results? Important details are missing, including the accurate description of the “graphene nanoplatelets” used. What is the isolation procedure used for the studied bacterial strains? What were the growth conditions (Petri dish, bottle, test-tube...). The lack of such details significantly impairs reproducibility, which is against the guidelines of this Journal.

We apologize for the lack of clarity. The lacking informations have been added in the discussion section. However, as reported by (Zanni et al., 2016), the advantage of using graphene nanoplatelets (GNPs) is their large lateral size together with the preferred growth orientation of the ZnO-NRs over the GNP surface. This shape contributes to increasing the adhesion of the nanostructures to the cell wall, enhancing the penetration of the ZnO-NRs through the cell membrane, hence killing cells through mechanical damage produced by the ZnO-NRs.

Moreover, details about the isolation protocols have been added in Materials and Methods section.

Zanni E, Chandraiahgari CR, De Bellis G, et al. Zinc Oxide Nanorods-Decorated Graphene Nanoplatelets: A Promising Antimicrobial Agent against the Cariogenic Bacterium Streptococcus mutans. Nanomaterials (Basel). 2016;6(10):179. Published 2016 Sep 29. doi:10.3390/nano6100179.

Since the goal of this work is to demonstrate the antibacterial and antibiofilm properties of their zinc oxide-based products, the authors must clarify their actual mechanism of action. Zinc oxide, as correctly mentioned in the Introduction, is known to be photocatalytic and thus able to generate reactive species which, in turn, can significantly contribute to the antimicrobial activity. The authors must precise whether they performed their tests in the dark or in presence of daylight, diffuse or direct, or under the light of fluorescent lightbulbs. Experimental controls in this sense should be performed as well. This is especially important by taking into account the final application of the proposed products, that is, on buildings exposed to sunlight. Only then, their hypothesis about antimicrobial activity by mechanical damaging could be accepted, or amended by including a photocatalytic component.

We thank the Reviewer and apologize for the missing information. We performed experiments with ZnO-NRs and ZNGs in room at 30°C under diffuse visible light. The information has been added in Materials and Methods section, where needed.

Why the authors decided not to show the results obtained with propidium iodide, that is, the dead cells? They are strongly encouraged to show them, at least in the Supporting Information.

We thank the Reviewer for the comment. We decided not to add the images reporting the biofilm staining with propidium iodide because they were characterized by the almost total absence of dead cells. The images obtained were almost entirely dark, apart from very few dead cells. For this reason we have decided to report the result it in the text, but if it is considered an important detail, we can enclose it in the Supporting Information.

Also: would the described products be significantly convenient (also from an economic perspective) compared to commercial ones? What about the possibility of deleterious effects on the stone themselves generated by side-reactions? A proper comparison is needed: the authors must extend the scarce discussion they provide at the end of the manuscript.

We thank the Reviewer for the comment. The discussion section has been extended accordingly.

In general, the writing style should be improved. Especially in the Introduction and Discussion sections, there are some ingenuities, including the repeated use of expressions such as “nanoknives” and so on, which does not sound professional.

We thank the Reviewer for the comment. The writing style has been improved.

The authors should also consider to rewrite the final sections of the Abstract and of the Conclusions, since they give the impression that the hazards and limitations refer to the products they propose, rather than to usual ones.

We apologize for the lack of clarity. The sections have been modified accordingly.

In Figure 5a, the scale is clearly different from that of the other ones: 100 microns compared to 20 microns. This makes very difficult, if not impossible, to properly compare the pictures. Also, in the legend, the authors should indicate more clearly the nature of the treatment used: “sprayed with 250 μg/mL” of what?

We thank the Reviewer for the comment. Figure 5 and legend have been modified accordingly.

Figure 7: the image of one bacteria flattened on a carpet of nanorods is not enough to provide convincing evidence for the “mechanical” antimicrobial mechanism. See before.

In general, by accompanying the SEM images with EDX maps would make the results more convincing, especially by showing the distribution of carbon (from bacteria) in samples treated with only zinc oxide.

We thank the Reviewer for the comment. We have changed Figure 7, where is better observable the mechanical activity of graphene-based nanomaterials. In particular, the penetration of ZnO-nanorods in the bacterial membrane is well visible.

It is also apparent, from the results showed, that the activity of the different materials tested is not the same for different bacteria strains: in other words, they cannot be considered for “general purpose”. This, in turn, means that they might favor the selection of one or more bacterial species, possibly exacerbating the stone degradation problems.

We thank the Reviewer for the comment. In this work, we compared two different ZnO-based nanomaterials, ZnO-nanorods and ZNGs. The proliferation of a bacterial strain is observed only when ZnO-nanorods were used. Using the ZNGs, instead, we obtained a good percentage of antimicrobial activity with all the tested bacteria highlighting that it could be a promising resource for general purposes.

What is the durability of the proposed materials? That is, for how long they will retain their antimicrobial properties, especially under normal weathering conditions, before a new application would be necessary?

The study concerned the evaluation of the effectiveness of the ZnO-nanomaterials on different substrates. Obviously, studies on their durability will be the subject of future researches. Indeed, weathering studies could be prepared in lab condition, but above all under normal outdoor conditions, with a minimum time of one year to evaluate the efficiency of the treatment. However, the efficacy of the products was checked at this stage even after two months, resulting still active.

The authors must provide a satisfying explanation to all these points in order to make their work, which anyway remains a proof-of-concept (the authors must clarify this point) publishable in Nanomaterials.

Minor details:

Lines 120, 126, 142 and so on: instead of “H2Odd”, it is better to say “deionized water”.

Line 123: it is “propidium”.

Line 140: correct the exponent.

The text has been modified accordingly.

Reviewer 2 Report

The manuscript under review deals with a frequented topic of the use of nanomaterials as means to suppress biodeterioration of historical artefacts. Generally, it reads as a detailed report on laboratory research. What is completely missing is the explanation or at least a suggestion of the explanation of the mechanism of the action of ZnO nanoparticles. Another issue is the difficult comparability of the data obtained using either ZnO-NR or ZNG. The concentration of ZNG presented in Fig. 6 of 250 microgram/mL is much higher than that used for ZnO-NR (see Figure 1: up to 100 microgram/mL). To sum up, the manuscript is of potential interest for the readership in the field of the development and use of nanoparticle biocides, however, a thorough revision is need to render it acceptable for the publication.

Author Response

The manuscript under review deals with a frequented topic of the use of nanomaterials as means to suppress biodeterioration of historical artefacts. Generally, it reads as a detailed report on laboratory research. What is completely missing is the explanation or at least a suggestion of the explanation of the mechanism of the action of ZnO nanoparticles. Another issue is the difficult comparability of the data obtained using either ZnO-NR or ZNG. The concentration of ZNG presented in Fig. 6 of 250 microgram/mL is much higher than that used for ZnO-NR (see Figure 1: up to 100 microgram/mL). To sum up, the manuscript is of potential interest for the readership in the field of the development and use of nanoparticle biocides, however, a thorough revision is need to render it acceptable for the publication.

We thank the Reviewer for the comment. The explanation of the mechanism of the action of ZnO nanoparticles has been added in the discussion section. About the different concentrations used in the experiments with stone speciments, we performed the tests using 250 microgram/mL, to better cover the porous surfaces of the stones. Indeed, the porosity of the bricks makes them easily penetrated by bacteria, which can escape the antimicrobial action of nanomaterials, allowing their proliferation. On the contrary, using ZnO-nanorods, a good percentage of antimicrobial activity is already reached with 100 microgram/mL. We have added the explanation in the text.

Reviewer 3 Report

Manuscript Title: Antibacterial effect of zinc oxide-based nanomaterials on environmental biodeteriogens affecting the historical buildings

Manuscript Type: Article
Presented paper fulfills the Journal Scopus. The research objectives and results are clearly stated. The authors motivated by cited literature carried out experiments and known knowledge. The title of the article reflects the research problem undertaken. The article is original. The presented paper is suitable for publication in Nanomaterials, after minor revision.

Comments :

Minor English correction is needed. In my opinion, in the article, some information on the studied materials are missing. In the description of the materials should nevertheless find a synthesis summary, not just the literature link itself. All used shortcuts need to explain/define. (e.g. H2Odd) Line 233 -It would be good if at least the authors refer to this data in the appendix or the specific publication. This itching asks for clarification.

Author Response

Manuscript Title: Antibacterial effect of zinc oxide-based nanomaterials on environmental biodeteriogens affecting the historical buildings

Manuscript Type: Article

Presented paper fulfills the Journal Scopus. The research objectives and results are clearly stated. The authors motivated by cited literature carried out experiments and known knowledge. The title of the article reflects the research problem undertaken. The article is original. The presented paper is suitable for publication in Nanomaterials, after minor revision.

Comments :

Minor English correction is needed. In my opinion, in the article, some information on the studied materials are missing. In the description of the materials should nevertheless find a synthesis summary, not just the literature link itself. All used shortcuts need to explain/define. (e.g. H2Odd) Line 233 -It would be good if at least the authors refer to this data in the appendix or the specific publication. This itching asks for clarification. 

We thank the Reviewer for the comment. The text and in particular Materials and Methods section have been modified accordingly. 

Round 2

Reviewer 1 Report

The authors amended the manuscript according to the suggestions. I would still recommend a language check and less emphasis regarding the synergistic effect of ZnO and GNP, unless more evidence could be provided.

Author Response

We thank the anonymous reviewer for improving our manuscript. In agreement with the indication, some sentences have been modified.